# Chemical Compositions of *Scutellaria baicalensis* Georgi. (Huangqin) Extracts and Their Effects on ACE2 Binding of SARS-CoV-2 Spike Protein, ACE2 Activity, and Free Radicals

**DOI:** 10.3390/ijms25042045

**Published:** 2024-02-07

**Authors:** Boyan Gao, Hanshu Zhu, Zhihao Liu, Xiaohua He, Jianghao Sun, Yanfang Li, Xianli Wu, Pamela Pehrsson, Yaqiong Zhang, Liangli Yu

**Affiliations:** 1Institute of Food and Nutraceutical Science, School of Agriculture and Biology, Shanghai Jiao Tong University, Shanghai 200240, China; gaoboyan@sjtu.edu.cn (B.G.); zhuhanshu@sjtu.edu.cn (H.Z.); 2Department of Nutrition and Food Science, University of Maryland, College Park, MD 20742, USA; 3Methods and Application of Food Composition Laboratory, Beltsville Human Nutrition Research Center, Agricultural Research Service, United States Department of Agriculture, Beltsville, MD 20705, USApamela.pehrsson@usda.gov (P.P.); 4Western Regional Research Center, Agricultural Research Service, United States Department of Agriculture, Albany, CA 94710, USA

**Keywords:** *Scutellaria baicalensis* Georgi., huangqin, COVID-19, SARS-CoV-2 spike protein, ACE2, chemical profile, antioxidant

## Abstract

The water and ethanol extracts of huangqin, the roots of *Scutellaria baicalensis* Georgi. with potential antiviral properties and antioxidant activities, were investigated for their chemical profiles and their abilities to interfere with the interaction between SARS-CoV-2 spike protein and ACE2, inhibiting ACE2 activity and scavenging free radicals. A total of 76 compounds were tentatively identified from the extracts. The water extract showed a greater inhibition on the interaction between SARS-CoV-2 spike protein and ACE2, but less inhibition on ACE2 activity than that of the ethanol extract on a per botanical weight concentration basis. The total phenolic content was 65.27 mg gallic acid equivalent (GAE)/g dry botanical and the scavenging capacities against HO^●^, DPPH^●^, and ABTS^●+^ were 1369.39, 334.37, and 533.66 µmol trolox equivalent (TE)/g dry botanical for the water extract, respectively. These values were greater than those of the ethanol extract, with a TPC of 20.34 mg GAE/g, and 217.17, 10.93, and 50.21 µmol TE/g against HO^●^, DPPH^●^, and ABTS^●+^, respectively. The results suggested the potential use of huangqin as a functional food ingredient in preventing COVID-19.

## 1. Introduction

The coronavirus disease 2019 (COVID-19), caused by severe acute respiratory syndrome coronavirus 2 (SARS-CoV-2) and its variants, leads to respiratory system injury and multi-organ lesions [1]. In the past few years, COVID-19 has swept the world and claimed many lives, causing a global catastrophic pandemic. According to WHO, as of 14 June 2023, COVID-19 has caused 767,984,989 confirmed cases globally, 6,943,390 of which resulted in loss of life [2]. The current approaches for COVID-19 prevention and therapy include different types of vaccines such as inactivated vaccines and mRNA vaccines, as well as antiviral drugs such as Remdesivir and Molnupiravir [3,4], but their distributions in the lungs restrict the efficacy of these drugs [5,6]. Also, these approaches may have possible side effects in some patients. For better efficacy and safety, effective preventive and therapeutic approaches besides vaccines and pharmaceuticals are still needed for SARS-CoV-2 and its mutants.

Research has revealed that SARS-CoV-2 enters the host cells via the binding of its spike (S) protein to the angiotensin-converting enzyme 2 (ACE2) of the host cells, the known SARS-CoV receptor [7,8,9]. In lung cells, SARS-CoV-2 uses the serine protease TMPRSS2 to initiate the receptor-binding domain (RBD) of its S-protein, binding to the ACE2 of the host cells and promoting entry; the virus subsequently releases RNA in the cytoplasm for replications, causing infection symptoms [9,10,11]. Furthermore, ACE2 inhibitors may reduce the availability of ACE2 for SARS-CoV-2 S-protein binding and have garnered attention as another potential means to reduce the risk of SARS-CoV-2 infection. In addition, SARS-CoV-2 infection causes an inflammatory response, the cytokine storm, which leads to an increased production of reactive oxygen species (ROS) and induces oxidative stress, causing cell damage [12]. The cytokine storm is a potentially deadly immune system response leading to inflammation and tissue damage throughout the body, which has been found in severe cases of COVID-19 [12,13]. Free radical scavengers might be important for calming down the infection-induced oxidative stress. Recent clinical trials have further confirmed that consuming antioxidants may expedite recovery from COVID-19 and potentially provide preventive effects against COVID-19 infection. Therefore, exploring suitable botanical samples could effectively enhance the conditions for preventing and treating COVID-19.

*Scutellaria baicalensis* Georgi. is a plant widely distributed in Northeast Asia. Its root is named huangqin and has been used for centuries in traditional Chinese medicine for its biological properties. Huangqin has been shown to have antitumor [14,15], antibacterial [16,17,18], and antiviral [19,20] activities in clinical studies. The flavonoids are considered its primary active components. Flavonoids have been shown to be a promising therapeutic intervention for several health problems in in silico, in vitro, in vivo, and clinical studies, as they act on different targets while producing less toxicity [21]. Scutellarein, baicalein, and oroxylin A are the flavonoid compounds detected in huangqin and have been tested and verified by molecular docking and in vitro studies with preventive effects against SARS-CoV-2 infection via inhibiting chymotrypsin-like protease (3CLpro) activity, which is responsible for the maturation of non-structural proteins in viruses [22,23]. Therefore, this study was conducted to investigate the chemical compositions of water and ethanol extracts of huangqin, their inhibitory effects on SARS-CoV-2 to ACE2 binding and ACE2 activity, and their free radical scavenging capacities. The results give information on the potential ability of huangqin to intervene in ACE2-mediated SARS-CoV-2 infection and provide a scientific foundation for the development of preventive and therapeutic agents from huangqin against COVID-19.

## 2. Results

### 2.1. Chemical Compositions

The tentative identification of 76 compounds in the huangqin water and ethanol extracts was carried out according to high-resolution full mass spectrometry, MS^2^ scanning, and literature data (Table 1 and Appendix A, Figure 1). All the compounds are flavonoids including 56 flavones (**11**, **13**–**15**, **18**, **22**, **24–30**, **32**, **34**–**39**, **41**–**76**), 14 flavanones (**3**, **6**, **7**, **9**, **10**, **12**, **17**, **19**–**21**, **23**, **31**, **33**, **40**), and 6 flavonols (**1**, **2**, **4**, **5**, **8**, **16**) (Table 1). All 76 compounds were detected in the water extract, while 71 of them were found in the ethanol extract (Figure 1). In the water extract, apigenin 7-*O*-glucuronide (**27**), baicalein (**62**), wogonin (**70**), oroxylin A-7-*O*-glucuronide (**44**), and wogonoside (**50**) were the five major compounds, while in the ethanol extract, baicalein (**62**), wogonin (**70**), skullcapflavone II (**73**), oroxylin A (**74**), and apigenin 7-*O*-glucuronide (**27**) were the five major compounds. Baicalein (**62**) and wogonin (**70**) were the two major compounds in both the water and ethanol extracts, which have been shown capable of binding to the SARS-CoV-2 chymotrypsin-like protease (3CLpro), a preferred target in SARS-CoV-2 treatment [23,24].

Peak #**34** with a retention time of 28.22 min was selected as an example (Table 1 and Figure 2) for tentative structure identification in this study. The molecular ion detected in the positive ion mode ([M + H]^+^) is *m*/*z* 447.09207, which is consistent with C_21_H_19_O_11_ (mass accuracy, −0.263 ppm). And the molecular ion in the negative ion mode ([M – H]^–^) is *m*/*z* 445.07694, which is consistent with C_21_H_17_O_11_ (mass accuracy, 0.904 ppm). These results indicate that the molecular formula of peak #**34** is C_21_H_18_O_11_. The major fragment ions detected in the positive mode are at *m*/*z* 429.0802 ([M + H–H_2_O]^+^), 285.0755 ([M + H–162]^+^), and 271.0599 ([M + H–glucuronide]^+^). The fragment ions of [M + H–162]^+^ are the result of a possible internal cleavage in the *O*-glucuronide group at its A ring. The major fragment ion detected in the negative mode is 269.0450 ([M – H–glucuronide]^–^). Based on these results and database information, this peak was tentatively identified as baicalin. Baicalin is a representative active flavone in huangqin, which has been proven to have antioxidant, antiviral, antitumor, and other biological effects [20,30,31,32,33].

### 2.2. Inhibition on SARS-CoV-2 Spike Protein and ACE2 Interaction

The interaction between ACE2 and SARS-CoV-2 spike protein is a crucial step in the path of COVID-19 replication and transmission. Therefore, the effectiveness of target extracts in inhibiting this interaction is an important indicator for treating COVID-19 or defending against viral transmission. The huangqin water and ethanol extracts were found to have inhibitory effects on the interaction between SARS-CoV-2 spike protein and ACE2 (Figure 3). Initially, the huangqin water extract was tested at a concentration of 100 mg dry botanical weight equivalent (DBWE). After being mixed with reactant solvent, the final concentration of the huangqin water extract was 33.3 mg DBWE/mL, and it caused a 76.86% inhibition of the interaction activity. Since the study did not allow for pure ethanol, samples of pure-ethanol-extracted huangqin (100 mg DBWE/mL ethanol) were diluted tenfold with water to prepare 10 mg DBWE/mL. After mixing with the reactant solvent, the ethanol extract (3.3 mg dry botanical equivalents/mL) showed a 34.65% inhibition. To compare the inhibition efficiency of the water and ethanol extracts of huangqin at the same concentration, both extracts were diluted by 20 times, resulting in a final concentration of 1.7 mg DBWE/mL. The results indicated that the 1.7 mg DBWE/mL water extract had no inhibitory effects, while the 1.7 mg DBWE/mL ethanol extract showed 12.89% inhibition. This indicates that the ethanol extract of huangqin has a higher inhibition efficiency against the interaction between SARS-CoV-2 spike protein and ACE2.

### 2.3. Inhibition on ACE2 Enzyme Activity

Both the huangqin water and ethanol extracts showed significant inhibition of ACE2 (Figure 4). The 33.3 mg dry botanical equivalents/mL and 3.3 mg dry botanical equivalents/mL water extracts inhibited 100.84% and 88.52% of ACE2 activity, respectively, while 33.3 and 3.3 mg dried botanical equivalent/mL ethanol extracts showed 68.43% and 5.97% inhibition, which were significantly weaker to the water extract of the same concentration.

### 2.4. Total Phenolic Contents (TPC) and Antioxidant Assays

The TPC values of the huangqin water and ethanol extracts were 65.27 and 20.34 mg GAE/g dry botanical, respectively (Figure 5). The results of our experiment were greater than those in previously reported data, where the TPC value of water extract was 3.85 mg GAE/g and that of ethanol extract was 3.65 mg GAE/g [34], probably due to the different extraction methods. Compared to previously reported extraction methods, which involved using water or ethanol at 37 °C and gently shaking for 2 h, the present study utilized an 85 °C water bath for 2 h, followed by keeping it at ambient temperature for an additional 22 h to achieve full extraction. The ethanol extract was also kept at ambient temperature for 24 h. It is widely acknowledged that both extraction temperature and duration can greatly influence extraction effectiveness. The results of this study demonstrate that using a relatively high temperature water extraction method is a more effective approach for extracting phenolic compounds from huangqin samples.

The in vitro antioxidant effect of the huangqin extracts was initially evaluated by three common free radical scavenging assays, including relative hydroxy radical scavenging capacity (HOSC), relative DPPH^●^ scavenging capacity (RDSC), and relative ABTS^●+^ scavenging capacity (ABTS). The free radical scavenging ability of the huangqin water and ethanol extracts are shown in Figure 6. The HOSC value of the huangqin water extract was as high as 1369.39 µmol TE/g, which was more than seven times higher than that of the ethanol extract (164.17 µmol TE/g). The RDSC and ABTS values of the water extract were 334.37 and 533.66 µmol TE/g, respectively, both significantly higher than those of ethanol extract (10.93 and 50.21 µmol TE/g, respectively) (*p* < 0.05).

## 3. Discussion

A total of 76 flavonoids were identified, and the kinds and relative intensity of the detected compounds showed some differences between different extractions. All the 76 compounds were found in water extract, but only 71 of them were detected in ethanol extract. The relative content of the same substances in water extract was higher than in ethanol extract. These differences imply differences in their biological activities.

ACE2 mediates the entry of SARS-CoV-2 into host cells as a receptor, so blocking the combination between ACE2 and SARS-CoV-2 or inhibiting ACE2 activity may reduce viral infection. The huangqin extracts showed their ability to interfere in ACE2-related infection, supported by our results of chemical compositions. Huangqin contains abundant flavonoids including scutellarein and baicalein, which have showed higher selectivity index (SI) in SARS-CoV-2 pseudovirus assay than cepharanthine, an inhibitor of viral entry [22]. The SI value is the ratio between the toxic concentration and effective biological activity concentration of a compound, and a high SI value represents the effectiveness and safety of a drug in preventing viral infection [35]. In addition, biolayer interferometry (BLI) and molecular docking studies confirmed that flavonoids can prevent viral infection by blocking the interaction of SARS-CoV-2 spike receptor binding domain (RBD) with ACE2 receptor [22]. Since the high infectivity of SARS-CoV-2 caused by its binding affinity to ACE2 (which is 10–20 times higher than that of SARS-CoV [36]), the inhibition of huangqin extracts and huangqin flavonoids against this binding is of great importance.

As for ACE2 activity, quercetin with 3′,4′-dihydroxylated structures and its metabolites inhibited recombinant human (rh) ACE2 activity in in vitro experiments [37]. The inhibitory effect of angiotensin-converting enzyme was significantly enhanced by structures with 3′,4′-dihydroxylation at the B ring and the C2 = C3 bond at the C ring of flavonoids, which are very common in compounds identified in huangqin [38]. However, ACE2 is also one of the key enzymes of the renin-angiotensin system (RAS), and its cleaving and modifying product angiotensin-(1-7) exerts vasodilation, anti-fibrosis, anti-proliferation, anti-inflammatory, and other functions through the combination with G-protein-coupled receptor MAS [10]. Inhibition of ACE2 may imbalance the RAS, which in turn causes other dysfunctions on the organism. Therefore, further research on the relationship between ACE2 activity and health condition are necessary.

Free radicals are associated with cytotoxic damage and inflammatory responses, and excessive free radical accumulation can lead to oxidative stress [12]. Oxidative stress has been found to play an important role in the development of various diseases, including cancer [39], neurodegenerative diseases [40], and cardiovascular diseases [41]. Oxidative stress is regarded to be a key factor in COVID-19, supported by the fact that COVID-19 patients exhibit higher levels of oxidative stress markers [42,43]. Increasing evidence showed that foods rich in antioxidants might have remarkable bioactivities in preventing or even therapying COVID-19 and/or its related symptoms, just like the therapeutic effects of antioxidants in patients with sepsis and acute lung injury. For example, applying glutamine or antioxidant vitamins as nutritional supplements or medications to critically septic patients may attenuate oxidative stress by increasing oxygenation rates, improving glutathione levels and enhancing immune response [12]. A pilot clinical trial carried in Spain verified that the administration of food supplements containing antioxidants and trace elements showed shorter hospital stays and early recovery after infected severe COVID-19 [44]. Another multi-center study revealed that the use of hydroxytyrosol, a polyphenol antioxidant, exhibited a statistically significant preventive effect against COVID-19 infection. [45]. Therefore, it is necessary to examine the antioxidant capacity of drug components in developing preventive and therapeutic approaches for COVID-19.

Phenolic compounds are one of the main classes of secondary metabolites in botanicals and are good antioxidants, acting as free radical terminators or metal chelators [46]. The total phenolic content (TPC) shows the antioxidant capacity of the tested material to some extent. In in vitro free radical scavenging assays, HOSC values suggest that huangqin extracts, especially water extracts, exhibited an excellent scavenging effect on hydroxyl radicals, which are the most reactive and detrimental ROS in biological systems [47]. The scavenging of initial free radicals, including hydroxyl radicals, prevents first-chain initiation and thus disrupts chain reactions to reduce oxidative stress [46]. However, the antiradical activity against stable radicals, including DPPH^●^ and ABTS^●+^, cannot fully represent the antioxidant activity of the test sample in biological systems. Therefore, the in vivo antioxidant activity of huangqin extracts needs to be further investigated. In general, the huangqin water extract had significantly stronger activity, indicating that water was a more suitable solvent than ethanol in huangqin’s bioactive extraction. This is reassuring because water extraction is easy, safe, and similar to the commonly used herbal tea preparation. Further studies will continue to prioritize the evaluation of specific bioactive compounds derived from huangqin. The aim is to elucidate the effects and mechanisms of these compounds through both in vitro and in vivo experiments.

## 4. Materials and Methods

### 4.1. Materials

Folin-Ciocalteu reagent (FC reagent), 6-hydroxy-2,5,7,8-tetramethylchroman-2-carboxylic acid (trolox), gallic acid, fluorescein (FL), 2,2-diphenyl-1-picrylhydrazyl (DPPH^∙^), 2,2′azinobis (3-ethylbenzothiazoline-6-sulfonic acid), diammonium salt (ABTS), ferric chloride (FeCl_3_), and hydrogen peroxide (H_2_O_2_) (30%) were purchased from Sigma-Aldrich (St. Louis, MO, USA). LC-MS-grade formic acid and acetonitrile were from Merck (Darmstadt, Germany). SARS-CoV-2 spike ACE2 interaction Inhibitor Screening Assay Kit (No. 502050) and ACE2 Inhibitor Screening Assay Kit (No. 502100) were from Cayman (Ann Arbor, MI, USA). All other chemicals used in this study were analytical grade and supplied by Fisher Scientific (Hampton, NH, USA) without further processing.

### 4.2. Sample Preparation and Extraction

Chinese Pharmacopeia grade commercial huangqin (*Scutellaria baicalensis* Georgi. Root) dry herbal decoction pieces were obtained from a local pharmacy located in Rockville, MD, USA. The huangqin herbal decoction pieces were pulverized into powder with a micromill grinder (Bel Art Products, Pequannock, NJ, USA), resulting in a particle size of less than 40 mesh. For both extraction methods, 5 g of huangqin powder was extracted using 50 mL of solvent. Ultrapure water was used to prepare the huangqin water extract, which was subjected to an 85 °C water bath for 2 h (with a ratio of 1:10, *w*/*v*) and then left at room temperature for 22 h to ensure complete extraction. The ethanol extract was obtained by extracting the huangqin powder with pure ethanol at room temperature for 24 h (with a ratio of 1:10, *w*/*v*). After extraction, the water and ethanol extracts were centrifuged, and the supernatants were collected without further filtration. The extracts were stored at −20 °C for subsequent analyses. Each milliliter of extract was considered equivalent to 0.1 g of dry huangqin. Triplicate samples were performed for both water and ethanol extractions.

### 4.3. Chemical Compositions of Huangqin (Scutellaria baicalensis Georgi. Root)

Chemical compositions of the huangqin water and ethanol extracts were identified according to our published protocol [48]. Chromatographic separation was performed on a Vanquish UHPLC (Thermo Fisher Scientific, Norristown, PA, USA) using an UltraShield pre-column (UltraShield, Santa Clara, CA, USA) and an Agilent Eclipse Plus-C18 UHPLC column (150 mm × 2.1 mm, 1.8 μm) (Agilent, Santa Clara, CA, USA), with an injection volume of 1 μL. Mobile phase A was 0.1% formic acid in water (*v*/*v*) and B was 0.1% formic acid in acetonitrile (*v*/*v*). The column was pre-equilibrated with 2% B for 5 min first. The gradient was programed as follows: 10% B in the first 15 min, 40% B during 15–35 min, 95% B during 35–55 min, and held for 5 min. Then the column was re-equilibrated with 2% B for 10 min. The flow rate was 0.3 mL/min.

The MS and MS^2^ data were obtained using an Orbitrap Fusion ID-X Tribrid mass spectrometer (Thermo Fisher Scientific, Norristown, PA, USA). The mass scan range was *m*/*z* 120 to 1200. The spray voltage was 3900 V in positive and 2500 V in negative ion modes. The temperatures of the ion transport tube and the vaporizer were 300 and 275 °C, respectively. The obtained data were processed using Xcalibur^TM^ (Thermo Fisher Scientific, Norristown, PA, USA).

### 4.4. Inhibitory Effects of Huangqin Extracts on SARS-CoV-2 Spike Protein and ACE2 Interaction

The inhibitory effects of the huangqin extracts on SARS-CoV-2 spike protein and ACE2 binding were measured using a SARS-CoV-2 Spike-ACE2 Interaction Inhibitor Screening Assay Kit (No. 502050). The procedure was performed according to the instructions of the product. After the reaction, the absorbance of the 96-well plate was read at 450 nm on Tecan M200 Pro microplate reader (Tecan Group Ltd., Mannedorf, Switzerland). The results were calculated based on Equation (1) and showed as the percent (%) inhibition, while *Abs_Sample_* represents the absorption of sample and the same is true for all of the rest. Pure water and pure ethanol were measured as negative control, and all the results presented were already the results of deducting negative control.
(1)$$\%\;inhibition=\frac{{Abs}_{Sample}-{Abs}_{Background}}{{Abs}_{100\%Initial}-{Abs}_{Background}}\times 100\%$$

### 4.5. Inhibitory Effects of Huangqin Extracts on ACE2

The inhibitory effects of the huangqin extracts on ACE2 enzyme activity were measured using a Cayman ACE2 Inhibitor Screening Assay Kit (No. 502100). According to the instructions of the product, 5 µL of solvent or huangqin extract sample solution, 75 µL of ACE2 assay buffer, and 10 µL of ACE2 enzyme were added to wells of a 96-well to react. After 30 min of incubation, the fluorescence (AF) (λ_ex_ = 320 nm, λ_em_ = 405 nm) of the plate was read by Tecan M200 Pro microplate reader (Tecan Group Ltd., Mannedorf, Switzerland). The results were calculated based on Equation (2). *AF_Sample_* stands for the fluorescence of samples, and the same is true for all of the rest. All the results about the direct inhibitory effects of the huangqin extracts on ACE2 were showed as the percent (%) inhibition.
(2)$$\%\;inhibition=(1-\frac{{AF}_{Sample}-{AF}_{Background}}{{AF}_{100\%Initial}-{AF}_{Background}})\times 100\%$$

### 4.6. Total Phenolic Content (TPC) Determination

Total phenolic content (TPC) of the huangqin water and ethanol extracts was determined according to laboratory protocol. A mixture of 3 mL of ultrapure water, 50 µL of solvent, standard gallic acid or huangqin extraction sample, and 250 µL of FC reagent were vortexed for 5 s in a test tube. After a few minutes, 750 µL of 20% Na_2_CO_3_ (*w*/*v*) was added and a two-hour reaction was started in the dark at room temperature. The absorbance of all samples was measured at 765 nm by multifunction microplate reader (Tecan M200 Pro, Tecan Group Ltd., Mannedorf, Switzerland). The TPC of the huangqin water and ethanol extracts was showed as milligram gallic acid equivalents per gram of huangqin sample (mg GAE/g).

### 4.7. Relative Hydroxy Radical Scavenging Capacity (HOSC)

Relative hydroxy radical scavenging capacity (HOSC) of the huangqin water and ethanol extracts were tested according to laboratory protocol. In a 96-well plate, 170 µL of working FL solution, 30 µL of blank, trolox standards or sample, 40 µL of H_2_O_2_ working solution, and 60 µL of FeCl_3_ working solution were added and shaken for 15 s. The fluorescence intensities at excitation wavelength of 485 nm and emission wavelength of 528 nm were measured every 5 min for 5 h by Tecan M200 Pro microplate reader (Tecan Group Ltd., Mannedorf, Switzerland). After calculating the area under the curve (AUC), the results were expressed as trolox equivalent per gram of huangqin sample (µmoles TE/g).

### 4.8. Relative DPPH^●^ Scavenging Capacity (RDSC)

Relative DPPH^●^ scavenging capacity (RDSC) of the huangqin water and ethanol extracts were tested according to laboratory protocol. Equivalent volume of solvent, trolox standards or sample, and 0.2 mM DPPH^●^ working solution were mixed in wells of a 96-well plate. The absorbance at 515 nm was measured every 60 s for 90 min by Tecan M200 Pro microplate reader (Tecan Group Ltd., Mannedorf, Switzerland). The AUCs were calculated and the results were reported as µmol trolox equivalent per gram of huangqin sample (µmoles TE/g).

### 4.9. Relative ABTS^●+^ Scavenging Capacity (ABTS)

Relative ABTS^●+^ scavenging capacity (ABTS) of the huangqin water and ethanol extracts were tested according to laboratory protocol. To start the reaction, 2 mL ABTS^●+^ working solution and 160 μL of trolox standards or sample was mixed and vortexed. The absorbance at 734 nm was measured at the 90th second of mixing by Genesys 20 visible spectrophotometer (Thermo Fisher Scientific, Norristown, PA, USA). The results were reported as µmol trolox equivalent per gram of huangqin sample (µmoles TE/g).

### 4.10. Statistical Analysis

All experiments were conducted in triplicate and the data were reported as the mean ± standard deviation (SD). The comparison between two data groups was examined via *t*-test by IBM SPSS Statistics (version 25.0, SPSS, Inc., Chicago, IL, USA), and *p*-values less than 0.05 were considered statistically significant and marked by different letters (a and b) in all figures. All the figures were charted by GraphPad Prism (version 8.0, Graphpad Software Inc., San Diego, CA, USA). The mass data and spectra were obtained and analyzed by Xcalibur^TM^ (Version 4.2, Thermo Fisher Scientific, Norristown, PA, USA).

## 5. Conclusions

In summary, this study presents an analysis of the chemical profiles of huangqin’s water and ethanol extracts, along with a preliminary assessment of their inhibitory effects on the binding of SARS-CoV-2 spike protein to ACE2 and ACE2 activity. Additionally, the study examines their scavenging ability against various free radicals. Using UHPLC-MS determination, a total of 76 flavonoid compounds were identified, suggesting the bioactive properties of huangqin. Among the identified flavonoid components, prominent flavone glycosides, such as baicalin, taxifolin 7-O-glucoside, kaempferol 3-O-glucuronide, carthamidin 7-O-glucuronide, and scutellarin, along with their aglycones, were found in significant amounts in huangqin extracts. Particularly, baicalin and scutellarin emerged as crucial flavonoid compounds, with previous results confirming the bioactive effects of baicalin in preventing COVID-19.

Notably, the study reveals that huangqin’s water extract exhibits a stronger inhibition of ACE2 enzyme activity but a weaker inhibition of the interaction between SARS-CoV-2 spike protein and ACE2 when compared to the ethanol extract. This observation aligns with the notion that water tends to extract more polar compounds from huangqin, resulting in higher levels of flavone glycosides and more potent COVID-19 inhibitory effects, and vice versa. Furthermore, the scavenging capacities of the water extract on HO^●^, DPPH^●^, and ABTS^●+^ were found to be superior to those of the ethanol extract, indicating that the water extract may possess enhanced antioxidant activity.

The results suggest huangqin as a promising candidate for inclusion in preventive and therapeutic drugs for COVID-19. However, differences in components and bioactivities between water and ethanol extracts were evident. Further in vitro and in vivo experiments, as well as clinical trials, are necessary to elucidate the mechanisms and confirm the systemic effects of huangqin.

## Figures and Tables

**Figure 1 ijms-25-02045-f001:**
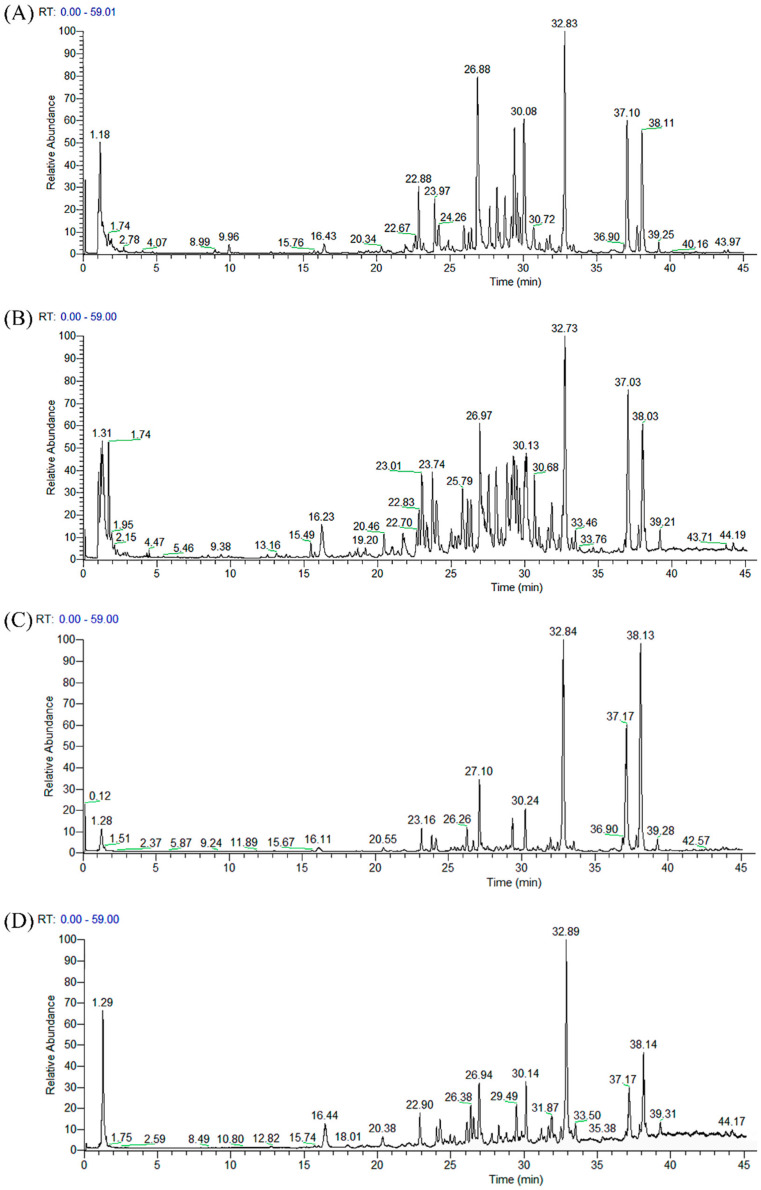
Chromatogram of huangqin water extract (WE) in positive (**A**) and negative (**B**) ionization modes, and huangqin ethanol extract (EE) in positive (**C**) and negative (**D**) ionization modes.

**Figure 2 ijms-25-02045-f002:**
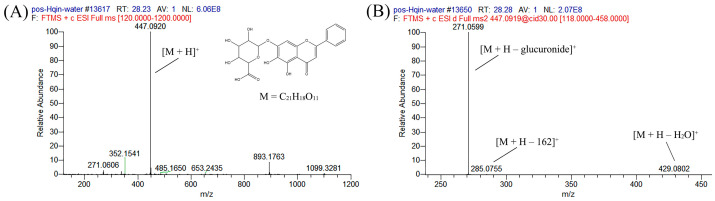
Identification of baicalin (compound **34**). Full scan MS (**A**) and MS^2^ (**B**) in positive ionization mode.

**Figure 3 ijms-25-02045-f003:**
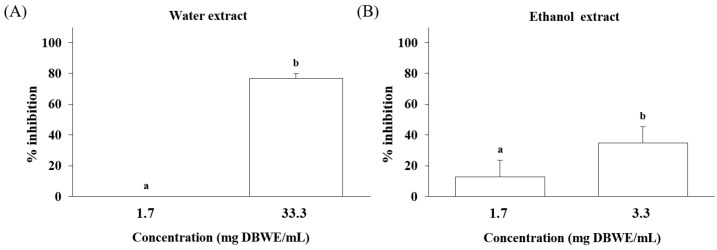
Inhibitory effects of the huangqin (**A**) water and (**B**) ethanol extracts on SARS-CoV-2 spike protein and ACE2 interaction. DBWE stands for dry botanical weight equivalents. Values are the mean ± SD of triplicate experiments. Letters a and b stand for significant differences (*p* < 0.05).

**Figure 4 ijms-25-02045-f004:**
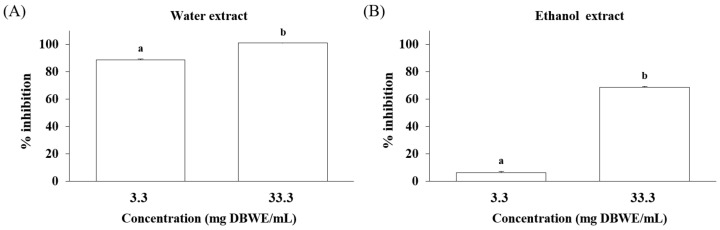
Inhibitory effects of the huangqin (**A**) water and (**B**) ethanol extracts on ACE2 activity. DBWE stands for dry botanical weight equivalents. Values are the mean ± SD of triplicate experiments. Letters a and b stand for significant differences (*p* < 0.05).

**Figure 5 ijms-25-02045-f005:**
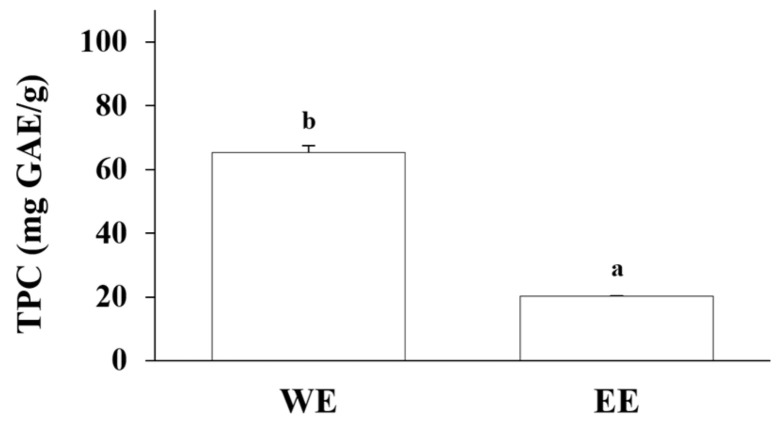
Total phenolic content (TPC) of the huangqin water (WE) and ethanol extracts (EE). Results are expressed as the mean ± SD of triplicate experiments and values are on a dry botanical weight basis. Letters a and b stand for significant differences (*p* < 0.05) between the water and ethanol extracts.

**Figure 6 ijms-25-02045-f006:**
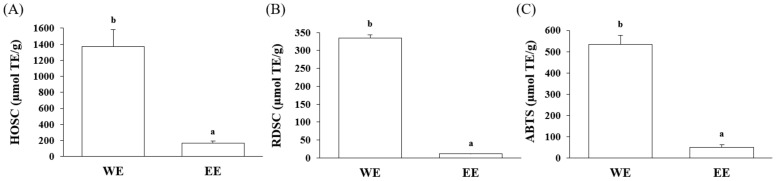
Relative free radical scavenging activities of the huangqin water (WE) and ethanol extracts (EE) against (**A**) HO^●^, (**B**) DPPH^●^, and (**C**) ABTS^●+^. Results are expressed as the mean ± SD of triplicate experiments and values are on a dry botanical weight basis. Letters a and b stand for significant differences (*p* < 0.05) between the water and ethanol extracts.

**Table 1 ijms-25-02045-t001:** Characterization of compounds identified in huangqin.

ID	Positive Mode (ESI^+^)				Negative Mode (ESI^−^)				Formula	Name	Ref.
	Retention Time	Exptl. [M + H]^+^	Fragment Ions	Mass Error (ppm)	Retention Time	Exptl. [M – H]^−^	Fragment Ions	Mass Error (ppm)			
1	12.81	467.11838	449.1074, **305.0652**, 287.0547	−0.048	13.16	465.10297	447.0916, **437.1072**, 339.0703, 285.0391, 241.0493, 177.0184	0.468	C_21_H_22_O_12_	Taxifolin 7-*O*-glucoside	[25]
2	13.10	463.08698	**445.0755**, 427.0645, 371.0753, 311.0396, 231.0281	−0.264	nd	nd	nd	nd	C_21_H_18_O_12_	Kaempferol 3-*O*-glucuronide	[25]
3	16.00	465.10254	447.0912, 345.0594, **303.0495**	−0.457	15.75	463.08752	435.0922, **301.0341**, 283.0237, 151.0030	0.902	C_21_H_20_O_12_	Carthamidin 7-*O*-glucuronide	[26]
4	16.43	305.06535	287.0548, 153.0182	−0.751	16.23	303.05042	285.0400, 125.0240	1.620	C_15_H_12_O_7_	Isomer of pentahydroxyflavanone	[25]
5	nd	nd	nd	nd	18.13	303.05026	285.9080	1.092	C_15_H_12_O_7_	Isomer of pentahydroxyflavanone	
6	19.59	481.09747	**305.0651**, 169.0128	−0.410	19.35	479.08258	**303.0502**, 285.0397	1.175	C_21_H_20_O_13_	5,6,7,3′,4′-Pentahydroxy flavanon 7-*O*-glucuronide	[25]
7	20.29	479.11853	461.0697, **317.0650**, 303.0495	0.266	20.47	477.10352	331.0298, **315.0500**	1.609	C_22_H_22_O_12_	5,7,2′-trihydroxy-6-methoxyflavanone 7-*O*-glucuronide	[25]
8	20.34	303.04993	285.0389, 127.0386	0.003	20.46	301.03470	283.0243, 257.0451, 193.0138, 151.0032, 125.0240	1.398	C_15_H_10_O_7_	Viscidulin I	[27]
9	21.98	465.10238	447.0919, 303.0494, 289.0704	−0.801	21.76	463.08755	445.0768, 287.0552, 269.0449, 193.0347	0.967	C_21_H_20_O_12_	Isocarthamidin 7-*O*-glucuronide	[26]
10	22.09	303.08609	**285.0752**, 257.0803	−0.741	21.94	301.07108	283.0611, 257.0820, **161.0241**, 139.0398	1.380	C_16_H_14_O_6_	Isomer of trihydroxy-methoxyflavanone	[27]
11	22.69	463.08679	301.0705, **287.0547**	−0.675	22.83	461.07183	299.0551, **285.0395**	0.819	C_21_H_18_O_12_	Scutellarin	[27]
12	22.83	465.10196	447.0918, 429.0812, 303.0861, **289.0702**	−1.704	22.97	463.08740	445.1133, **287.0552**, 269.0448, 193.0346	0.643	C_21_H_20_O_12_	Eriodictyol 7-*O*-glucuronide	[26]
13	22.89	549.15970	**531.1490**, 513.1387, 483.1283, 429.1176, 411.1072	−1.033	23.06	547.14534	529.1346, 487.1239, **457.1131**, 427.1026, 367.0813, 337.0708	1.321	C_26_H_28_O_13_	Chrysin 6-*C*-arabinoside-8-*C*-glucoside	[25]
14	23.97	549.15974	**531.1488**, 513.1385, 483.1280, 429.1176, 411.1072	−0.960	23.74	547.14531	529.1346, 487.1239, **457.1131**, 427.1026, 367.0813, 337.0708	1.266	C_26_H_28_O_13_	Chrysin 6-*C*-glucoside-8-*C*-arabinoside	[25]
15	24.20	287.05499	269.0439, 237.0388, 219.0283 153.0177, 137.0229, 107.0487	−0.085	24.05	285.04013	267.0292, 217.0499, 199.0394, 151.0030, 133.0290, 107.0134	2.686	C_15_H_10_O_6_	5,7,2′,6′-tetrahydroxyflavone	[25]
16	24.29	303.04980	285.0390, 229.0493, 195.0286	−0.426	24.12	301.03486	273.0402, 229.0503,151.0034	1.930	C_15_H_10_O_7_	Isomer of pentahydroxyflavone	
17	24.56	479.11810	**317.0650**, 303.0858	−0.631	24.42	477.10312	**301.0708**, 286.0470	0.368	C_22_H_22_O_12_	5,7,2′-Trihydroxy-8-methoxy flavanone 7-*O*-glucuronide	[25]
18	24.90	417.11777	**399.1070**, 381.0966, 351.0861, 297.0755, 255.0649	−0.572	25.04	415.10266	397.0925, 337.0708, 325.0707, **295.0602**, 267.0657, 253.0498,	0.726	C_21_H_20_O_9_	Isomer of chrysin 8-*C*-glucoside	[27]
19	25.19	289.07056	271.0599, 127.0388	−0.362	25.27	287.05568	269.0446, 125.0238	2.318	C_15_H_12_O_6_	Carthamidin	[25]
20	25.23	479.11826	461.1073, **303.0859**	−0.297	25.36	477.10318	301.0707	0.896	C_22_H_22_O_12_	Isomer of trihydroxy-methoxyflavanone *O*-glucuronide	[25]
21	25.47	303.08606	**285.0754**, 257.0806	−0.840	25.49	301.07092	**161.0242**, 139.0399	0.848	C_16_H_14_O_6_	Isomer of trihydroxy-methoxyflavanone	[27]
22	25.95	477.10254	459.0920, **301.0707**, 286.0479	−0.445	25.76	475.08755	**299.0547**, 284.0315	0.942	C_22_H_20_O_12_	5,6,7-trihydroxy-8-methoxy-7-*O*-glucuronide	[25]
23	26.02	289.07061	271.0598	−0.189	25.87	287.05562	269.0457	2.109	C_15_H_12_O_6_	Isocarthamidin	[25]
24	26.10	463.12326	**301.0707**, 287.0550	−0.492	25.92	461.10843	**299.0551**, 285.0396	1.284	C_22_H_22_O_11_	5,7,2′-trihydroxy-6-methoxyflavone 7-*O*-glucoside	[25]
25	26.29	347.07605	332.0525, 314.0421	−0.270	26.11	345.06076	330.0373, 315.0141	0.771	C_17_H_14_O_8_	Viscidulin III	[27]
26	26.48	287.05490	269.0441, 241.0493, 169.0130, 119.0490	−0.399	26.37	285.04016	267.0295, 239.0346, 137.0240, 117.0346	2.791	C_15_H_10_O_6_	Scutellarein	[25]
27	26.88	447.09209	429.1021, 313.0900, **271.0602**	−0.219	26.97	445.07682	**269.0449**, 175.0242	0.634	C_21_H_18_O_11_	Apigenin 7-*O*-glucuronide	[28]
28	27.10	433.11291	271.0598	−0.031	27.17	431.09761	413.0876, **269.0451**	0.781	C_21_H_20_O_10_	Apigenin 7-*O*-glucoside	[25]
29	27.32	417.11766	**399.1078**, 381.0972, 351.0865, 297.0760	−0.836	27.31	415.10263	397.0918, 337.0705, 325.0706, **295.0601**	0.654	C_21_H_20_O_9_	Isomer of chrysin 8-*C*-glucoside	[27]
30	27.59	463.12305	**301.0704**, 287.0547	−0.946	27.41	461.10815	**443.0602**, 299.0547	0.677	C_22_H_22_O_11_	Isomer of trihydroxy methoxyflavone *O*-glucoside	
31	27.72	449.10745	431.0968, 413.0864, 395.0760, 327.0346, **273.0755**, 169.0136	−0.864	27.57	447.09262	429.0816, **271.0605**, 243.0655	0.967	C_21_H_20_O_11_	Dihydrobaicalin	[25]
32	27.93	477.10263	**301.0702**, 286.0480	−0.246	27.75	475.08746	**299.0547**, 284.0322	0.753	C_22_H_20_O_12_	5,7,8-trihydroxy-6-methoxy flavone-7-*O*-glucuronide	[25]
33	28.14	479.11813	461.1084, **303.0863**	−0.569	27.98	477.10336	301.0711	1.273	C_22_H_22_O_12_	Isomer of trihydroxy-methoxyflavanone *O*-glucuronide	
34	28.22	447.09207	429.0802, 285.0755, **271.0599**	−0.263	28.09	445.07694	269.0450	0.904	C_21_H_18_O_11_	Baicalin	[28]
35	28.51	417.11801	399.1061, 351.0861, 297.0748, **255.0648**	0.003	28.57	415.10275	397.0924, 295.0607, **253.0502**	0.943	C_21_H_20_O_9_	Chrysin 6-*C*-glucoside	[27]
36	28.62	317.06558	302.0419, 153.0181	0.003	28.68	315.05048	300.0270, 283.0245, 151.0035	1.749	C_16_H_12_O_7_	Pedalitin	[25]
37	28.72	447.09203	285.0755, **271.0598**	−0.353	28.77	445.07694	**269.0450**, 175.0243	0.904	C_21_H_18_O_11_	Norwogonin 7-*O*-glucuronide	[28]
38	28.76	477.10239	**301.0702**, 286.0480	−0.760	28.84	475.08768	**299.0551**, 284.0320	1.216	C_22_H_20_O_12_	5,7,2′-trihydroxy-6-methoxy flavone 7-*O*-glucuronoide	[25]
39	28.78	301.07053	**286.0471**, 167.0334	−0.447	28.89	299.05563	284.0318, 271.0606, 212.0472	2.058	C_16_H_12_O_6_	4′-hydroxywogonin	[29]
40	28.92	463.12326	**301.0703**, 286.0477	−0.922	28.97	461.10846	446.0840, **299.0549**	1.349	C_22_H_22_O_11_	(2S)-5,7-Dihydroxy-6-methoxyflavanone 7-*O*-glucuronide	[27]
41	28.95	433.11258	415.1751, **271.0598**, 255.0648	−0.792	28.99	431.09790	**269.0445**, 253.0496	1.454	C_21_H_20_O_10_	Baicalein 7-*O*-glucoside	[27]
42	29.21	431.09689	**255.0649**, 238.0605, 146.3212	−0.889	29.13	429.08185	**253.0501**, 175.0242	0.529	C_21_H_18_O_10_	Chrysin 7-*O*-glucuronide	[25]
43	29.25	361.09169	346.0679, 331.0446, 328.0574, 313.0340	−0.288	nd	nd	nd	nd	C_18_H_16_O_8_	Isomer of trihydroxy-trimethoxyflavone	
44	29.40	461.10787	299.0910, 285.0754, **271.0597**	0.070	29.23	459.09261	**283.0601**, 268.0370	0.920	C_22_H_20_O_11_	Oroxylin A-7-*O*-glucuronide	[25]
45	29.62	477.10275	**301.0703**, 286.0473	−0.005	29.48	475.08780	**299.0551**, 284.0318	1.468	C_22_H_20_O_12_	Isomer of trihydroxy methoxy flavone *O*-glucuronoide	[25]
46	29.76	287.05469	269.0444, 153.0181, 137.0233	−1.131	29.61	285.04013	241.0500, 151.0032	2.686	C_15_H_10_O_6_	Isoscutellarein	[25]
47	29.81	447.09201	429.0816, 285.0757, **271.0601**	−0.398	29.69	445.07700	427.0663, 401.0869, **269.0450**, 251.0345	1.039	C_21_H_18_O_11_	Baicalein 6-*O*-glucuronide	[25]
48	29.91	331.08102	**316.0572**, 298.0467, 287.0546, 197.0442	−0.632	29.81	329.06564	**314.0421**, 299.0188, 195.0291	0.185	C_17_H_14_O_7_	Isomer of trihydroxy dimethoxyflavone	[27]
49	30.05	433.11276	**271.0597**	−0.377	nd	nd	nd	nd	C_21_H_20_O_10_	Isomer of dihydroxyflavanone *O*-glucoside	
50	30.08	461.10784	**285.0753**, 271.0591	−0.005	30.08	459.09262	**283.0602**, 268.0370, 175.0241	0.941	C_22_H_20_O_11_	Wogonoside	[25]
51	30.40	347.07605	332.0522, 317.0287, 314.0417	−0.270	30.44	345.06094	330.0364, 315.0134	1.293	C_17_H_14_O_8_	Isomer of tetrahydroxy-dimethoxyflavone	
52	30.72	491.11816	**315.0858**, 300.0634	−0.494	30.68	489.10320	**313.0706**, 175.0244	0.915	C_23_H_22_O_12_	5,7-dihydroxy-8,2′-dimethoxyflavone 7-*O*-glucuronide	[27]
53	31.11	301.07053	**286.0469**, 255.0651, 121.0282	−0.447	30.98	299.05530	**284.0317**, 137.0239, 117.0350	0.955	C_16_H_12_O_6_	Hispidulin	[29]
54	31.32	361.09128	346.0677, 331.0444, **328.0573**, 313.0338, 227.0547, 212.0311	−1.423	31.20	359.07629	**344.0527**, 329.0293, 326.0425, 254.9857, 225.0396, 210.0164	0.407	C_18_H_16_O_8_	5,2′,5′-trihydroxy-6,7,8-trimethoxyflavone	[27]
55	31.57	331.08096	**316.0574**, 301.0340, 298.0469	−0.813	31.53	329.06607	**314.0424**, 299.0192	1.492	C_17_H_14_O_7_	Isomer of trihydroxy dimethoxyflavone	[27]
56	31.61	301.07058	**286.0465**, 283.0465, 255.0646	−0.281	31.60	299.05551	**284.0323**, 281.0456, 117.0353	1.657	C_16_H_12_O_6_	Isomer of trihydroxy-methoxyflavone	
57	31.84	271.06006	253.0494, 225.0544	−0.147	31.85	269.04507	241.0501, 225.0552, 171.0447	2.305	C_15_H_10_O_5_	Apigenin	[25]
58	31.93	331.08111	**316.0574**, 301.0340, 298.0469	−0.360	31.96	329.06617	**314.0424**, 299.0198	1.765	C_17_H_14_O_7_	Viscidulin II	[27]
59	32.45	331.08115	**316.0576**, 298.0471, 270.0523, 183.0287, 169.0131	−0.239	32.35	329.06613	**314.0424**, 191.0344, 137.0239	1.674	C_17_H_14_O_7_	5,7,6′-trihydroxy-8,2′-dimethoxyflavone	[27]
60	32.66	361.09137	**346.0679**, 331.0445, 328.0574	−1.174	32.55	359.07651	**344.0519**, 329.0282, 254.9849	1.020	C_18_H_16_O_8_	Isomer of trihydroxy-trimethoxyflavone	[27]
61	32.70	301.07047	**286.0469**, 283.0600, 255.0648, 105.0333	−0.646	32.59	299.05557	**284.0321**, 153.0190	1.857	C_16_H_12_O_6_	Tenaxin II	[27]
62	32.83	271.05975	253.0493, 225.0544, 197.0596	−1.291	32.73	269.04501	251.0344, 241.0501, 223.0396, 195.0447, 169.0654	2.082	C_15_H_10_O_5_	Baicalein	[25]
63	32.83	287.05493	269.0441, 243.0650, 225.0544	−0.294	nd	nd	nd	nd	C_15_H_10_O_6_	Isomer of tetrahydroxyflavone	[25]
64	33.21	331.08099	**316.0575**, 301.0341, 298.0470	−0.723	33.22	329.06583	**314.0425**, 299.0197	0.762	C_17_H_14_O_7_	5,8,2′-trihydroxy-6,7-dimethoxyflavone	[27]
65	33.44	301.07056	**286.0469**, 283.0601, 255.0648, 183.0288	−0.347	33.46	299.05569	**284.0322**, 255.0661, 212.0476, 165.9915, 110.0015	2.259	C_16_H_12_O_6_	5,6,7-trihydroxy-4′-methoxyflavone	[25,27]
66	33.82	331.08090	316.0574, 301.0339, **298.0468**	−0.994	33.76	329.06591	**314.0424**, 299.0190	1.005	C_17_H_14_O_7_	5,7,2′-trihydroxy-8,6′-dimethoxyflavone	[25]
67	35.32	331.08102	**316.0573**, 298.0468, 287.0547, 270.0521, 197.0442	−0.632	35.23	329.06595	**314.0418**, 299.0186, 285.0397, 268.9837	1.127	C_17_H_14_O_7_	Isomer of trihydroxy dimethoxyflavone	[27]
68	36.43	361.09219	**346.0680**, 331.0446, 328.0573, 313.0341	1.097	36.41	359.07550	**344.0522**, 329.0289, 326.0419	−1.793	C_18_H_16_O_8_	Isomer of trihydroxy-trimethoxyflavone	
69	36.90	345.09689	**330.0730**, 270.6946	0.031	36.81	343.08152	**328.0576**, 313.0345, 237.0397, 195.0293, 180.0058, 164.9828	0.847	C_18_H_16_O_7_	Skullcapflavone	[25]
70	37.09	285.07547	**270.0520**, 239.0702, 105.0334	−0.982	37.03	283.06080	**268.0370**, 163.0041, 110.0013	2.473	C_16_H_12_O_5_	Wogonin	[25]
71	37.18	255.06493	209.0595, 171.0287	−1.001	37.12	253.04984	**209.0602**, 143.0497, 107.0134	1.204	C_15_H_10_O_4_	Chrysin	[25]
72	37.78	315.08622	**300.0649**, 285.0414	−0.300	37.73	313.07117	**298.0475**, 283.0243, 180.0060	1.614	C_17_H_14_O_6_	5,8-dihydroxy-6,7-dimethoxyflavone	[25]
73	38.09	375.10721	**360.0836**, 345.0599, 327.0495, 227.0548	−0.624	38.02	373.09218	**358.0685**, 343.0451, 303.0506, 194.9932	1.035	C_19_H_18_O_8_	Skullcapflavone II	[25]
74	38.14	285.07561	**270.0518**, 239.0699	−0.491	38.10	283.06076	**268.0373**, 239.0710	2.332	C_16_H_12_O_5_	Oroxylin A	[25]
75	38.26	315.08625	**300.0624**, 271.0599	−0.205	38.22	313.07117	**298.0475**, 283.0243, 180.0063	1.614	C_17_H_14_O_6_	5,7-dihydroxy-6,8-dimethoxyflavone	[25]
76	39.26	345.09689	**330.0734**, 315.0499, 284.0679, 227.0549	0.031	39.21	343.08151	**328.0580**, 313.0348, 282.0532, 269.0454	0.818	C_18_H_16_O_7_	Tenaxin I	[25]

Exptl.: [M + H]^+^ represents experimental *m*/*z* of molecular ion in positive mode; [M – H]^−^ represents experimental *m*/*z* of molecular ion in negative mode; Ref. represents references; WE—water extracts of huangqin; EE—ethanol extracts of huangqin; nd represents not detectable. The bolded molecular weight suggests that this fragment possesses the highest natural abundance among the compound.

## Data Availability

The original contributions presented in the study are included in the article/Appendix A, further inquiries can be directed to the corresponding author.

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
