# Peer review of "Chemical Compositions of Scutellaria baicalensis Georgi. (Huangqin) Extracts and Their Effects on ACE2 Binding of SARS-CoV-2 Spike Protein, ACE2 Activity, and Free Radicals"

_ijms, 2024, doi:10.3390/ijms25042045_

Round 1

Reviewer 1 Report (New Reviewer)

Comments and Suggestions for Authors

In figures 3-6, error bars for EE group is very small (hardly can be seen). How could this possible with only triplicate experiments?

In this paper, it is stated that the results of their experiment were greater than those previously reported before, where the TPC value of water extract was 3.85 mg 145 GAE/g and that of ethanol extract was 3.65 mg GAE/g [29]. But you have not discussed the reason of this discrepancy. I would like to see how authors justify such a difference.

Since this paper is submitted to IJMS and deals with molecular domains, I am wondering whether including a schematic picture like “Structure of the SARS-CoV-2 spike receptor-binding domain bound to the ACE2 receptor within the paper, see Nature volume 581, pages215–220 (2020)” could be helpful? A schematic picture on how these identified extract (for instance Compound 34) can help us in combating Covid-19 may be more helpful.

Also, lines 202-204, authors state that “At present, antioxidant 202 therapy has not been tested in COVID-19, but antioxidants have demonstrated their therapeutic effects in patients with sepsis and acute lung injury.”. This is not true, because just only in pubmed, 65 clinical trial papers are indexed (1/19/2024), investigating the effect of supplementary antioxidants in Covid-19 patients. This section need to be corrected and discussed again. Similar rule applies to your Introduction section. Authors need to add some studies which show the effectiveness of antioxidants in COVID-19 diseases…SEE https://pubmed.ncbi.nlm.nih.gov/?term=antioxidant+covid-19&filter=pubt.clinicaltrial&filter=hum_ani.humans&sort=date&size=200

Minor points:

Lines 40-41: “The current approaches for COVID-19 prevention and therapy 40 include vaccines and antiviral drugs such as Remdesivir and Molnupiravir, respectively 41 [3, 4],”. Please note that Remdesivir and Molnupiravir are classified as drugs not as vaccines.

Line 102 it is stated that “Compound 34 was selected as an example (Table 1 and Figure 2) for tentative structure identification in this study.”. Then in lines 111-112, you state that “Based on these results and database 111 information, compound 34 was tentatively identified as baicalin.” Please elaborate further on this paragraph, as readers may find this as a confusing paragraph. Compound 34 was selected as an example? Was this a random understanding or it was clear for you from very beginning.

Author Response

Comment 1: In figures 3-6, error bars for EE group is very small (hardly can be seen). How could this possible with only triplicate experiments?

Answer: Thank you for your comment. We appreciate your feedback. Upon reviewing the figures, we acknowledge that there was an error in our previous statement. The error bars for EE in figure 3 are visually readable, so we assume you meant to refer to figures 4-6 instead.

Figure 4 presents the direct inhibition of extracts to ACE2 protein, and the corresponding values were determined using the Cayman ACE2 Inhibitor Screening Assay Kit, which is a commercially available kit. Figures 5 and 6 display the total phenolic contents and antioxidant activities of Huangqin extracts. These assays were conducted following a lab protocol that has been established in our own lab for over 20 years. These well-established in vitro analytical methods have demonstrated excellent reproducibility as long as they are performed under stable conditions. We have double-checked our triplicated raw data, and they have been confirmed to be correct. No changes have been made to the data.

Comment 2: In this paper, it is stated that the results of their experiment were greater than those previously reported before, where the TPC value of water extract was 3.85 mg GAE/g and that of ethanol extract was 3.65 mg GAE/g [29]. But you have not discussed the reason of this discrepancy. I would like to see how authors justify such a difference.

Answer: Different extraction methods might be the major reason induced such different TPC results. Discussion has been added about these differences (Line 161-168): Compared to previously reported extraction methods, which involved using water or ethanol at 37 ℃ and gently shaking for 2 hours, the present study utilized an 85 ℃ water bath for 2 hours, followed by keeping it at ambient temperature for an additional 22 hours to achieve full extraction. The ethanol extract was also kept at ambient temperature for 24 hours. It is widely acknowledged that both extraction temperature and duration can greatly influence extraction effectiveness. The results of this study demonstrate that using a relatively high temperature water extraction method is a more effective approach for extracting phenolic compounds from Huangqin samples.

Comment 3: Since this paper is submitted to IJMS and deals with molecular domains, I am wondering whether including a schematic picture like “Structure of the SARS-CoV-2 spike receptor-binding domain bound to the ACE2 receptor within the paper, see Nature volume 581, pages215–220 (2020)” could be helpful? A schematic picture on how these identified extracts (for instance Compound 34) can help us in combating Covid-19 may be more helpful.

Answer: Appreciate the comment. We really want to prepare a schematic picture to elucidate the bioactive compound like baicalin and its inhibiting COVID-19 effects, but based on our present study no DIRECT evidence could be used to prove the bioactive effects of specific compound in Huangqin extracts, since the aim of present study is to confirm the effectiveness of Huangqin extracts in inhibiting COVID-19 and its related symptoms. We might furtherly dig deeper working mechanisms of specific compounds, baicalin, taxifolin 7-O-glucoside, kaempferol 3-O-glucuronide, carthamidin 7-O-glucuronide, and scutellarin, along with their aglycones both in vitro and in vivo in our next study. This fact has been added and discussed in the conclusion section (Line 362-385):

In summary, this study presents an analysis of the chemical profiles of Huangqin water and ethanol extracts, along with a preliminary assessment of their inhibitory effects on the binding of SARS-CoV-2 spike protein to ACE2 and ACE2 activity. Additionally, the study examines their scavenging ability against various free radicals. Using UHPLC-MS determination, a total of 76 flavonoid compounds were identified, suggesting the bioactive properties of Huangqin.

Among the identified flavonoid components, prominent flavone glycosides, such as baicalin, taxifolin 7-O-glucoside, kaempferol 3-O-glucuronide, carthamidin 7-O-glucuronide, and scutellarin, along with their aglycones, were found in significant amounts in Huangqin extracts. Particularly, baicalin and scutellarin emerged as crucial flavonoid compounds, with previous results confirming the bioactive effects of baicalin in preventing COVID-19.

Notably, the study reveals that Huangqin water extract exhibits stronger inhibition of ACE2 enzyme activity but weaker inhibition of the interaction between SARS-CoV-2 spike protein and ACE2 when compared to the ethanol extract. This observation aligns with the notion that water tends to extract more polar compounds from Huangqin, resulting in higher levels of flavone glycosides and more potent COVID-19 inhibitory effects, and vice versa.

Furthermore, the scavenging capacities of the water extract on HO, DPPH, and ABTS●+ were found to be superior to those of the ethanol extract, indicating that the water extract may possess enhanced antioxidant activity. The results suggest Huangqin as a promising candidate for inclusion in preventive and therapeutic drugs for COVID-19.

However, differences in components and bioactivities between water and ethanol extracts were evident. Further in vitro and in vivo experiments, as well as clinical trials, are necessary to elucidate the mechanisms and confirm the systemic effects of Huangqin.

Comment 4: Also, lines 202-204, authors state that “At present, antioxidant therapy has not been tested in COVID-19, but antioxidants have demonstrated their therapeutic effects in patients with sepsis and acute lung injury.”. This is not true, because just only in pubmed, 65 clinical trial papers are indexed (1/19/2024), investigating the effect of supplementary antioxidants in Covid-19 patients. This section needs to be corrected and discussed again. Similar rule applies to your Introduction section. Authors need to add some studies which show the effectiveness of antioxidants in COVID-19 diseases…SEE https://pubmed.ncbi.nlm.nih.gov/?term=antioxidant+covid-19&filter=pubt.clinicaltrial&filter=hum_ani.humans&sort=date&size=200

Answer: The sentence has been revised to ‘Increasing evidence showed that foods rich in antioxidants might have remarkable bioactivities in preventing or even therapying COVID-19 and/or its related symptoms, just like the therapeutic effects of antioxidants in patients with sepsis and acute lung injury.’ (Line 222-226)

And two more references have been added to elucidate the effectiveness of antioxidants in preventing COVID-19. ‘A pilot clinical trial carried in Spain verified that the administration of food supplement contains antioxidants and trace elements showed shorter hospital stay, early recovery after infected severe COVID-19 [39]. Another multi-center study revealed that the use of hydroxytyrosol, a polyphenol antioxidant, exhibited a statistically significant preventive effect against COVID-19 infection. [40].’ (Line 229-234).

And the effects of antioxidants in preventing and treating COVID-19 have also been added to the Introduction section: ‘Recent clinical trials have further confirmed that consuming antioxidants may expedite the recovery from COVID-19 and potentially provide preventive effects against COVID-19 infection. Therefore, exploring suitable botanical samples could effectively enhance the conditions for preventing and treating COVID-19.’ (Line 60-64)

Minor comment 1: Lines 40-41: “The current approaches for COVID-19 prevention and therapy 40 include vaccines and antiviral drugs such as Remdesivir and Molnupiravir, respectively 41 [3, 4],”. Please note that Remdesivir and Molnupiravir are classified as drugs not as vaccines.

Answer: The whole sentence has been revised to ‘The current approaches for COVID-19 prevention and therapy include different types of vaccines like inactivated vaccine and mRNA vaccines, as well as antiviral drugs such as Remdesivir and Molnupiravir’ to avoid misunderstanding (Line 40-42).

Minor comment 2: Line 102 it is stated that “Compound 34 was selected as an example (Table 1 and Figure 2) for tentative structure identification in this study.”. Then in lines 111-112, you state that “Based on these results and database 111 information, compound 34 was tentatively identified as baicalin.” Please elaborate further on this paragraph, as readers may find this as a confusing paragraph. Compound 34 was selected as an example? Was this a random understanding or it was clear for you from very beginning.

Answer: We just use one example to elucidate the compound identification progress. To avoid misunderstanding, the sentence has been revised to ‘Peak #34 with retention time at 28.22 min was selected as an example, …this peak was tentatively identified as baicalin’ (Line 107, 112, 117).

Reviewer 2 Report (New Reviewer)

Comments and Suggestions for Authors

Comments and Suggestions for Authors

 The article entitled: Chemical Compositions of Scutellaria baicalensis Georgi. (Huangqin) Extracts, Their Effects on SARS-CoV-2 Spike Protein to ACE2 Binding, ACE2 Activity, and Free Radicals is scientifically sound, and interesting on the effect of Huangqin on COVID-19 via SARS-CoV-2 spike protein to ACE2 binding, ACE2 activity, and free radicals. However, the authors have to fulfill information and discussion in some remarks below in order to clarify your review article for better understanding of the readers.

Remarks:

-   The author’s name is not in italic form for scientific name, it should be Scutellaria baicalensis Georgi. ### please correct through your MS.

-   Abstract: I think the author should re-write your abstract because difficult to understand and some words or some sentences have to be corrected i.e.

:    dry botanical??? botanical weight???

:    The water extract showed a greater inhibition on the interaction between 22 SARS-CoV-2 spike protein and ACE2, but less inhibition on ACE2 activity than that of the ethanol 23 extract on a per botanical weight concentration basis.???

     I would suggest to move Table 1 in this MS to Supplementary file and substitute with Table S1 (by adding one column of Ref in Table S1 before move to MS).

-   I would suggest also to move Figure S1 from Supplementary file to combine in Figure 1 of this MS in order to compare the chromatogram of water and ethanol extracts.

-   Page 9 of 16, 2.2. ### please re-writing because this paragraph was not clarified and difficult to understand.

-   Page 12 of 16, 4.2…… The water and ethanol extracts were centrifuged and the supernatants were taken and stored at -20 ℃ for analyses. ### Why did you not evaporate the solvent out of your extract before examination? How to make sure that the chemical components in the extracts are stable in the solvent? should explain!

-   Page 12-13 of 16, Line 272 and 281 ### I think the authors should write Equation 1, 2 and should explain all abbreviations i.e. AbsSample, AbsBackground, etc.

-   I think some data are missing in this MS, it will be good if in this MS exhibits the content (%) of each component in each extract, which will be good for discussion between the biological activity and outstanding compound in the extract.

Cheers,

Date of this review

21 January 2024

Comments on the Quality of English Language

Moderate editing of English language required.

Some sentences are difficult to understand.

Author Response

General comment: The article entitled: “Chemical Compositions of Scutellaria baicalensis Georgi. (Huangqin) Extracts, Their Effects on SARS-CoV-2 Spike Protein to ACE2 Binding, ACE2 Activity, and Free Radicals” is scientifically sound, and interesting on the effect of Huangqin on COVID-19 via SARS-CoV-2 spike protein to ACE2 binding, ACE2 activity, and free radicals. However, the authors have to fulfill information and discussion in some remarks below in order to clarify your review article for better understanding of the readers.

Answer: We sincerely thank the comments and questions from reviewer, and will try our best to answer all the questions and revise the manuscript based on the comments.

Comment 1: The author’s name is not in italic form for scientific name, it should be Scutellaria baicalensis Georgi. ### please correct through your MS.

Answer: The scientific name of Huangqin, Scutellaria baicalensis Georgi. has been corrected through the whole manuscript.

Comment 2: Abstract: I think the author should re-write your abstract because difficult to understand and some words or some sentences have to be corrected i.e. dry botanical??? botanical weight??? The water extract showed a greater inhibition on the interaction between  SARS-CoV-2 spike protein and ACE2, but less inhibition on ACE2 activity than that of the ethanol  extract on a per botanical weight concentration basis.???

Answer: The misunderstanding might result from the inaccurate descriptions in the sections of bioactivities and sample preparations, and these sections have been re-written to made it more clear (Line 124-141, 265-278).

Comment 3:   I would suggest to move Table 1 in this MS to Supplementary file and substitute with Table S1 (by adding one column of Ref in Table S1 before move to MS)

Answer: The positions of Table 1 and S1 in the manuscript and supplementary materials have been reversed, and the references have been added to the current Table 1 (Line 100).

Comment 4: I would suggest also to move Figure S1 from Supplementary file to combine in Figure 1 of this MS in order to compare the chromatogram of water and ethanol extracts.

Answer: Figure S1 has been moved to manuscript as Figure 1C and 1D to conveniently compare the chromatograms of water and ethanol extracts.

Comment 5: Page 9 of 16, 2.2. ### please re-writing because this paragraph was not clarified and difficult to understand.

Answer: Section 2.2 has been re-written to elucidate the exact concentrations of water and ethanol extracts of huangqin, and made comparasion between these two extracts in inhibiting the interaction of SARS-CoV-2 spike protein and ACE2.

Here is the revised paragraph. ‘The interaction between ACE2 and SARS-CoV-2 spike protein is a crucial step in the path of COVID-19 replication and transmission. Therefore, the effectiveness of target ex-tracts in inhibiting this interaction is an important indicator for treating COVID-19 or de-fending against viral transmission. Huangqin water and ethanol extracts were found to have inhibitory effects on the interaction between SARS-CoV-2 spike protein and ACE2 (Figure 3). Initially, the Huangqin water extract was tested at a concentration of 100 mg dry botanical weight equivalent (DBWE). After being mixed with reactant solvent, the final concentration of the Huangqin water extract was 33.3 mg DBWE/mL, and it caused a 76.86% inhibition of the interaction activity. Since the study did not allow for pure ethanol, sam-ples of pure ethanol extracted Huangqin (100 mg DBWE/mL ethanol) were diluted tenfold with water to prepare 10 mg DBWE/mL. After mixing with reactant solvent, the ethanol extract (3.3 mg dry botanical equivalents/mL) showed a 34.65% inhibition. To compare the inhibition efficiency of the water and ethanol extracts of Huangqin at the same concentra-tion, both extracts were diluted by 20 times, resulting in a final concentration of 1.7 mg DBWE/mL. The results indicated that the 1.7 mg DBWE/mL water extract had no inhibito-ry effects, while the 1.7 mg DBWE/mL ethanol extract showed 12.89% inhibition. This in-dicates that the ethanol extract of Huangqin has a higher inhibition efficiency against the interaction between SARS-CoV-2 spike protein and ACE2.’ (Line 124-141)

Comment 6: Page 12 of 16, 4.2…… The water and ethanol extracts were centrifuged and the supernatants were taken and stored at -20 ℃ for analyses. ### Why did you not evaporate the solvent out of your extract before examination? How to make sure that the chemical components in the extracts are stable in the solvent? should explain!

Answer: The whole 4.2 section about sample preparation and extraction has been re-written. ‘Chinese Pharmacopeia grade commercial huangqin (Scutellaria baicalensis Georgi. root) dry herbal decoction pieces were obtained from a local pharmacy located in Rockville, MD, USA. The huangqin herbal decoction pieces were pulverized into powder with a micromill grinder (Bel Art Products, Pequannock, NJ, USA), resulting in a particle size of less than 40 mesh. For both extraction methods, 5 g of huangqin powder was extracted using 50 mL of solvent. Ultrapure water was used to prepare the Huangqin water extract, which was sub-jected to an 85 ℃ water bath for 2 hours (with a ratio of 1:10, w/v) and then left at room temperature for 22 hours to ensure complete extraction. The ethanol extract was obtained by extracting the huangqin powder with pure ethanol at room temperature for 24 hours (with a ratio of 1:10, w/v). After extraction, the water and ethanol extracts were centrifuged, and the supernatants were collected without further filtration. The extracts were stored at -20 ℃ for subsequent analyses. Each milliliter of extract was considered equivalent to 0.1 g of dry huangqin. Triplicate samples were performed for both water and ethanol extractions.’ (Line 265-278).

Comment 7: Page 12-13 of 16, Line 272 and 281 ### I think the authors should write Equation 1, 2 and should explain all abbreviations i.e. AbsSample, AbsBackground, etc.

Answer: Equation 1 and 2 have been added to Line 306 and 317, and the parts of section 4.4 and 4.5 related with the equations have been revised.

The results were calculated based on equation 1 and showed as the percent (%) inhibition, while AbsSample represents the absorption of sample and the same is true for all of the rest. (Line 301-303)

The results were calculated based on equation 2. AFSample stands for the fluorescence of samples, and the same is true for all of the rest. All the results about the direct inhibitory effects of Huangqin extracts to ACE2 were showed as the percent (%) inhibition. (Line 314-316)

Comment 8: I think some data are missing in this MS, it will be good if in this MS exhibits the content (%) of each component in each extract, which will be good for discussion between the biological activity and outstanding compound in the extract.

Answer: Appreciate the comment. We’d really want to quantify every identified components, and directly compare their concentrations and possible bioactive contributions. But the major aim of present study is to elucidate the chemical profiles of Huangqin and investigate the appropriate extraction approaches against COVID-19. We re-written the conclusion section, add one paragraph about specific bioactive compounds in Huangqin ‘Among the identified flavonoid components, prominent flavone glycosides, such as baicalin, taxifolin 7-O-glucoside, kaempferol 3-O-glucuronide, carthamidin 7-O-glucuronide, and scutellarin, along with their aglycones, were found in significant amounts in Huangqin extracts. Particularly, baicalin and scutellarin emerged as crucial flavonoid compounds, with previous results confirming the bioactive effects of baicalin in preventing COVID-19.’ (Line 367-372), and propose future studies about specific bioactivity to COVID-19, ‘The results suggest Huangqin as a promising candidate for inclusion in preventive and therapeutic drugs for COVID-19. However, differences in components and bioactivi-ties between water and ethanol extracts were evident. Further in vitro and in vivo experi-ments, as well as clinical trials, are necessary to elucidate the mechanisms and confirm the systemic effects of Huangqin.’ (Line 381-385).

Reviewer 3 Report (New Reviewer)

Comments and Suggestions for Authors

ijms-2845069

Article Title:  Chemical Compositions of Scutellaria baicalensis Georgi. (Huangqin) Extracts, Their Effects on SARS-CoV-2 Spike Protein to ACE2 Binding, ACE2 Activity, and Free Radicals

Chemical profiles of the water and ethanol extracts of the Huangqin obtained from Scutellaria baicalensis Georgi roots were assayed by their abilities in interfering the interaction between SARS-CoV-2 spike protein and ACE2, inhibiting ACE2 activity and scavenging free radicals.

The work shows on potential ability of Huangqin to intervene in ACE2-mediated SARS-CoV- 2 infection, and suggest that can be used for the development of preventive and therapeutic agents against COVID-19, and 76 compounds were identificated in the Huangqin water and ethanol extracts by high-resolution full mass spectrometry

Comments

1)       Sample preparation and extraction

For both extracts, please indicate, in addition to the time and temperature, how the procedure was carried out, in relation to whether they were filtered, made up to the original volume, were they lyophilized and resuspended for the tests, it is necessary that the authors indicate how they standardized the extracts, as well as well as indicating the percentage yield obtained from the sample (dry sample? or fresh sample?)

2)       In point 2.2. Inhibition on SARS-CoV-2 spike protein and ACE2 interaction. Regard the follow sentence: “The 33.3 mg dry botanical equivalents/mL water extracts inhibited 76.86% of the interaction activity”, is necessary explain as was the mentioned previously, the extract preparation or standardization, this will make it possible to clarify the exact concentration used to each assay, and that the manuscript’s results can be reproducible based on the details given in the methods section.

3)       acronym DBWE,  Please, explain or define

 The results provide an advancement of the current knowledge of the Huangqin medicinal use against Covid-19.  The article is good  written, the data and analyses presented in an appropriate way ,  however is necessary respond on standardization of the sample’s extracts, that included to all assays (anti-Covid 10 and antioxidants) that are evaluated.

Author Response

Comment 1: The work shows on potential ability of Huangqin to intervene in ACE2-mediated SARS-CoV- 2 infection, and suggest that can be used for the development of preventive and therapeutic agents against COVID-19, and 76 compounds were identificated in the Huangqin water and ethanol extracts by high-resolution full mass spectrometry

1) Sample preparation and extraction

For both extracts, please indicate, in addition to the time and temperature, how the procedure was carried out, in relation to whether they were filtered, made up to the original volume, were they lyophilized and resuspended for the tests, it is necessary that the authors indicate how they standardized the extracts, as well as well as indicating the percentage yield obtained from the sample (dry sample? or fresh sample?)

Answer: The whole 4.2 section about sample preparation and extraction has been re-written. ‘Chinese Pharmacopeia grade commercial huangqin (Scutellaria baicalensis Georgi. root) dry herbal decoction pieces were obtained from a local pharmacy located in Rockville, MD, USA. The huangqin herbal decoction pieces were pulverized into powder with a micromill grinder (Bel Art Products, Pequannock, NJ, USA), resulting in a particle size of less than 40 mesh. For both extraction methods, 5 g of huangqin powder was extracted using 50 mL of solvent. Ultrapure water was used to prepare the Huangqin water extract, which was sub-jected to an 85 ℃ water bath for 2 hours (with a ratio of 1:10, w/v) and then left at room temperature for 22 hours to ensure complete extraction. The ethanol extract was obtained by extracting the huangqin powder with pure ethanol at room temperature for 24 hours (with a ratio of 1:10, w/v). After extraction, the water and ethanol extracts were centrifuged, and the supernatants were collected without further filtration. The extracts were stored at -20 ℃ for subsequent analyses. Each milliliter of extract was considered equivalent to 0.1 g of dry huangqin. Triplicate samples were performed for both water and ethanol extractions.’ (Line 265-278).

Comment 2: In point 2.2. Inhibition on SARS-CoV-2 spike protein and ACE2 interaction. Regard the follow sentence: “The 33.3 mg dry botanical equivalents/mL water extracts inhibited 76.86% of the interaction activity”, is necessary explain as was the mentioned previously, the extract preparation or standardization, this will make it possible to clarify the exact concentration used to each assay, and that the manuscript’s results can be reproducible based on the details given in the methods section.

Answer: We completely re-written the sample preparation section to clarify that ‘Each milliliter of extract was considered equivalent to 0.1 g of dry huangqin’ (Line 276-277), and the original extractions were then diluted for further bioactivity analyses. No change was made.

Comment 3: acronym DBWE,  Please, explain or define

The results provide an advancement of the current knowledge of the Huangqin medicinal use against Covid-19.  The article is good written, the data and analyses presented in an appropriate way ,  however is necessary respond on standardization of the sample’s extracts, that included to all assays (anti-Covid 19 and antioxidants) that are evaluated.

Answer: DBWE stands for dry weight equivalents (Line 130, 148-149).

And the sample preparation section has been completely re-written, each original extraction was considered 0.1 g dry huangqin per mL (Line 265-278). We sincerely appreciate reviewer’s comments, and carefully revised and checked the whole manuscript based on the comments.

Round 2

Reviewer 3 Report (New Reviewer)

Comments and Suggestions for Authors

ijms-2845069-peer-review-v2

Article Title:  Chemical Compositions of Scutellaria baicalensis Georgi. (Huangqin) Extracts, Their Effects on SARS-CoV-2 Spike Protein to ACE2 Binding, ACE2 Activity, and Free Radicals

Chemical profiles of the water and ethanol extracts of the Huangqin obtained from Scutellaria baicalensis Georgi roots were assayed by their abilities in interfering the interaction between SARS-CoV-2 spike protein and ACE2, inhibiting ACE2 activity and scavenging free radicals.

In this article was investigate the chemical compositions of Huangqin water and ethanol extracts, their inhibitory effects on SARS-CoV-2 to ACE2 binding and ACE2 activity, and their free radical scavenging potential

The authors reviewed and response that questions that were required,

Now is possible accept according to the review V2.

This manuscript is a resubmission of an earlier submission. The following is a list of the peer review reports and author responses from that submission.

Round 1

Reviewer 1 Report

Comments and Suggestions for Authors

The manuscript entitled ‘Chemical Compositions of Scutellaria baicalensis Georgi. (Huangqin) Extracts, Their Effects on SARS-CoV-2 Spike Protein to ACE2 Binding, ACE2 Activity, and Free Radicals’ by Boyan Gao and coauthors describes chemical profiles of the Huangqin extracts, the potential antiviral properties and antioxidant activities of Huangqin, and how the water and ethanol extracts of Huangqin interfere with the interaction between SARS-CoV-2 spike protein and ACE2 and inhibit ACE2 using commercial biochemical kits.

The paper suffers from some scientific problems and a few additional works are required.

Major comments:

It is not clear why in figure 3 different concentrations were used for water extract (1.7 and 33.3) and ethanol extract (1.7 and 3.3)?

Authors stated, ‘At the same concentration (1.7 mg dry botanical equivalents/mL in testing mixture), the ethanol extract showed higher inhibitory activity (EE1.7, 12.89% inhibition) than the water extract (WE1.7, not detected), suggesting that ethanol as solvent might have better inhibitory effect on SARS-CoV-2 spike protein and ACE2 interaction.’ However, inhibition of 12.89% or anything less than 65% is not considered notable. The authors stated the ethanol extract study demonstrated dose dependent effect; however, % inhibitions are too low and are rather questionable to compare. Please justify.

There are no proper negative controls. Particularly sterile distilled water and ethanol should be used as negative controls at the concentrations they present in the final test solution.

The results showed that none of the tested effects was comparative to the used positive controls (reference drug) and this should not be explained as effectiveness as described.

In Figure 3, 4, 5 and 6, mention the p values (a and b) properly in the figure legends as well as in the method section. Footnotes can be much clearer, and the legends used in the figure should be mentioned properly.

Chromatogram/ GC-MS analysis of phytocomponents of Huangqin ethanol extract should be included in the supplementary information.

Some more methods to measure antioxidant activity can be added like ferric reducing antioxidant power assay (FRAP)/oxygen radical absorbance capacity (ORAC) assay, etc.

Studies needed to identify the specific active biomolecule out of 71/76 flavonoids in the extracts that contribute to their antioxidant and antiviral properties by inhibiting ACE2 enzyme activity?

In vitro experiments are needed to validate the biological activity of the extract.

Minor comments:

In the abstract add a period (.) after Scutellaria baicalensis Georgi

Font size in all the figures is not uniform.

Reviewer 2 Report

Comments and Suggestions for Authors

ijms-2539717

The authors investigated aqueous and ethanolic extracts of Huangqin, roots of Scutellaria baicalensis Georgi with potential antiviral properties and antioxidant activities, for their ability to interfere with the interaction between SARS-CoV-2 spike protein and ACE2 by inhibiting ACE2 activity and scavenging free radicals . As a result: the aqueous extracts showed greater inhibition of the interaction between SARS-CoV-2 spike protein and ACE2, less inhibition of ACE2 activity by ethanol extract and good scavenging ability towards HO●, DPPH● and ABTS●, in preventing COVID19.

The manuscript is well- written and in a standart Endglish with need of slide modification.

Table 1 - contains too much overlapping data. Please model it for greater clarity.

The chemical structures can be separated in a second table and the most active compounds can be given a different color. The text below the table should conform to the font of the manuscript;

Figure 3 and 4 - formulas to be removed; are already given in the text

Figure 6. The resolution needs to be increased. All figures lack statistical significance marked with (*) or (**); Is significance P<0.05 everywhere?

Discussion: Commenting on the levels of free-radical neutralization and neutralization of oxidative changes by a given agent needs to be described in more detail. There are relevant articles in the literature that could supplement the manuscript.

Comments on the Quality of English Language

Minor editing of English language required

Round 2

Reviewer 1 Report

Comments and Suggestions for Authors

All previously raised questions and concerns are not sufficiently addressed.